# A Moderate Intake of Beer Improves Metabolic Dysfunction-Associated Steatotic Liver Disease (MASLD) in a High-Fat Diet (HFD)-Induced Mouse Model

**DOI:** 10.3390/molecules29245954

**Published:** 2024-12-17

**Authors:** Andrea Vornoli, Aymen Souid, Barbara Lazzari, Federica Turri, Flavia Pizzi, Emilia Bramanti, Beatrice Campanella, Cheherazade Trouki, Andrea Raffaelli, Marta Wójcik, Clara Maria Della Croce, Lucia Giorgetti, Vincenzo Longo, Emanuele Capra, Luisa Pozzo

**Affiliations:** 1Institute of Agricultural Biology and Biotechnology, National Research Council, Via Moruzzi 1, 56124 Pisa, Italy; andrea.vornoli@cnr.it (A.V.); souid.ayman@gmail.com (A.S.); andrea1.raffaelli@santannapisa.it (A.R.); lucia.giorgetti@ibba.cnr.it (L.G.); vincenzo.longo@ibba.cnr.it (V.L.); 2Institute of Agricultural Biology and Biotechnology, National Research Council, Via Corti 12, 20133 Milan, Italy; barbara.lazzari@ibba.cnr.it; 3Institute of Agricultural Biology and Biotechnology, National Research Council, Via dell’Università 6, 26900 Lodi, Italy; federica.turri@ibba.cnr.it (F.T.); flavia.pizzi@ibba.cnr.it (F.P.); 4Institute of Chemistry of Organometallic Compounds, National Research Council, Via Moruzzi 1, 56124 Pisa, Italy; emilia.bramanti@pi.iccom.cnr.it (E.B.); beatrice.campanella@pi.iccom.cnr.it (B.C.); 5Institute for Chemical and Physical Processes, National Research Council, Via Moruzzi 1, 56124 Pisa, Italy; cheherazade.trouki@pi.ipcf.cnr.it; 6Department of Pharmacy, University of Pisa, Via Bonanno 6, 56126 Pisa, Italy; 7Crop Science Research Center, Scuola Superiore Sant’Anna, Piazza Martiri della Libertà 33, 56127 Pisa, Italy; 8Sub-Department of Pathophysiology, Department of Preclinical of Veterinary Sciences, Faculty of Veterinary Medicine, University of Life Sciences in Lublin, Akademicka 12, 20-033 Lublin, Poland; marta.wojcik@up.lublin.pl

**Keywords:** steatosis, beer, gene expression, DNA methylation, cecal metabolites, phenolic compounds

## Abstract

Beer and its components show potential for reducing hepatic steatosis in rodent models through multiple mechanisms. This study aimed to evaluate beer’s anti-steatotic effects in a high-fat diet (HFD)-induced mouse model of Metabolic dysfunction-Associated Liver Disease (MASLD) and to explore the underlying mechanisms. In the HFD group, steatosis was confirmed by altered blood parameters, weight gain, elevated liver lipid content, and histological changes. These markers were normalized in the HFD+beer group, reaching levels similar to the control (CTR) group. Protein carbonylation and lipid peroxidation levels were consistent across all groups, suggesting that the model represents an early stage of MASLD without oxidative stress. Transcriptomic and CpG methylation analyses revealed clear distinctions between the CTR and HFD groups. RNA sequencing identified 162 differentially expressed genes (DEGs) between the CTR and HFD groups, primarily related to inflammation and lipid regulation. Beer consumption modified the health of the HFD mice, affecting inflammation but not lipid homeostasis (CTR vs. HFD+beer, DEGs = 43). The CpG methylation analysis indicated that beer lowered methylation, impacting genes linked to lipid accumulation and inflammation. A cecal metabolite analysis suggested that beer improved short-chain fatty acid metabolism (SCFA). In summary, a moderate beer intake may mitigate MASLD by modulating lipid metabolism and SCFA pathways, likely through polyphenol activity.

## 1. Introduction

Beer constitutes a widely consumed fermented beverage, representing 34.3% of the overall global alcohol consumption in 2016 [1]. Beer serves as a significant carrier of polyphenols, which, together with bitter acids, constitute beer’s antioxidants. Most of these compounds originate from malt, with only approximately 20% being derived from hops [2]. Consuming beer in low-to-moderate amounts has been shown to lower the risk of heart disease, compared to both non-drinkers and heavy drinkers [3]. This suggests that beer might have some heart-protective benefits, likely due to its polyphenol content. The exact mechanisms behind these potential benefits are not yet fully understood, but it is thought to be related to the antioxidant, anti-inflammatory, and lipid-regulating properties of the polyphenols and bitter acids in beer [4]. However, the impact of beer on liver health remains unclear. There is a lack of epidemiological studies examining the relationship between beer consumption and liver health, and the findings remain inconclusive, primarily because they are often contradictory [5,6,7]. In general, it has long been known that excessive alcohol consumption (e.g., intake > 50 g/day) is clearly linked to the development of liver steatosis [8]. Conversely, more recent population-based research indicates that a moderate alcohol intake (<20 g on 1–3 days per week) may reduce the likelihood of developing Metabolic dysfunction-Associated Liver Disease (MASLD) [9,10]. These findings align with other studies reporting that moderate drinkers (<20 g/day) exhibit a lower risk of diagnosis with non-alcoholic steatohepatitis (NASH) and fibrosis compared to lifetime abstainers [11]. Fermented alcoholic beverages, such as wine or beer, consumed in moderation can be seen as complex beverages. This is because the low intake of ethanol may pose a minimal risk, and the health benefits of the bioactive compounds within these beverages could counterbalance any potential negative effects. During digestion, fermentation in the gut provides energy for microbial growth and produces beneficial metabolites like short-chain fatty acids (SCFAs), amino acids, and vitamins. These compounds play key roles in regulating inflammation and stimulating the release of intestinal hormones within the body [12]. Emerging evidence suggests that polyphenols may play a crucial role in mitigating steatosis by modulating lipid metabolism, oxidative stress, and inflammation pathways [13]. Our laboratories have demonstrated that various polyphenol-rich substances of natural origin, such as brassicas and halophytes, can effectively improve steatosis across multiple fatty liver disease models [14,15]. The molecular characterization of liver alterations in HFD mouse models provided deeper insights into the evolution and progression of liver steatosis. The first experiments explored mRNA changes during hepatic adaptation to HFDs by identifying a switch from hepatic inflammatory response transitioning to steatosis, alongside the activation of genes related to lipids and triglycerides accumulation [16,17]. Together with transcriptomic variation, dynamic changes in the DNA methylation status of genes related to lipid metabolism and hepatic steatosis in mice by HFD-induced obesity were observed [18,19]. Both transcriptomic and epigenetic profiling of liver tissue were observed to reveal early signatures associated with hepatic disease transition and they were recently used to distinguish different stages of hepatic fibrosis and MASLD in mice and humans [20,21,22]. Transcriptomic changes in mice following HFD treatment were found to be reversible, with partial reversion occurring after weight loss [23], in other case mice with an anti-steatotic treatments showed a reversion of the HFD transcriptomic signature, although the extent varied depending on the treatment [24]. In the present study, the objectives were first to study the beer for its phenolic compound profile and its antioxidant capacity through in vitro tests, and then to verify the ability of a moderate daily dose of beer administered to mice to counteract the onset of diet-induced MASLD. At the mechanistic level, the study was corroborated by the analysis of gene expression and DNA methylation. Furthermore, cecal metabolomic changes were profiled to uncover the potential mechanisms underlying preventive effects of moderate daily beer consumption on MASLD.

## 2. Results

### 2.1. Bioactive Compounds and Antioxidant Capacity of Beer

The beer displayed total polyphenol content of 25.01 ± 1.27 mg GAE/100 mL, a flavonoids content of 3.17 ± 0.17 mg CE/100 mL, and a flavonols content of 3.07 ± 0.23 mg QE/100 mL (Table 1).

The beer’s antioxidant potential was evaluated using three in vitro tests measuring antioxidant capacity (ORAC), metal-related antioxidant power (FRAP), and radical scavenging activity (DPPH). Our findings revealed that the beer demonstrated significant antioxidant activity, with an ORAC value of 103.10 ± 7.01 mg TE/100 mL, a FRAP value of 39.11 ± 0.42 mg TE/100 mL, and a DPPH of 154.77 ± 13.05 mg TE/100 mL (Table 1).

### 2.2. Quantification of Phenolic Compounds in Beer by UHPLC-ESI-MS/MS

To gain a more thorough insight into the phenolic composition of the beer, we employed UHPLC-ESI-MS/MS to elucidate the principal constituents and their respective quantities. Out of the twenty-seven phenolic compounds identified, eight were identified as phenolic acids, while the remaining nineteen were classified as flavonoids, as outlined in Table 2.

Chlorogenic, vanillic, and gallic acids were identified as the most prevalent phenolic acids. Among the identified flavonoid compounds, we found that flavonols were the most abundant in our beer, with a notable predominance of 3-O-quercetin derivatives as the primary flavonoid aglycones. Specifically, quercetin 3-O-glucoside (isoquercetin) emerged as the predominant polyphenol (14,684.60 ± 2678.60 µg/100 mL). With an average concentration of 2245.90 µg/100 mL, the phenylethanoid tyrosol represents the second-most abundant phenolic compound found in our beer.

### 2.3. In Vivo Experimental Study

#### 2.3.1. The Influence of Beer Consumption on Serum Biochemical Parameters and Body Weight

The biochemical analysis investigated differences in hepatotoxicity and lipid profiles markers among the four experimental groups through a serum analysis. As compared to the controls, the HFD group showed elevated levels of aspartate aminotransferase (AST) and alanine aminotransferase (ALT), suggesting possible liver damage. The HFD+beer group exhibited significantly reduced transaminases levels compared to the HFD group, indicating a potential mitigating effect of the moderate beer consumption. Both TC and LDL-C levels significantly increased in the HFD group compared to the control group. The concurrent daily administration of beer to hyperlipidemic mice led to a decrease in these levels, with the reduction in LDL-C being statistically significant. Plasma glucose levels showed a significant increase with both the administration of beer alone and with the administration of an HFD. However, mice in the HFD group exhibited significantly higher blood glucose levels compared to those in the CTR+beer group. Conversely, when the mice were simultaneously administered an HFD and beer, their blood glucose levels returned to those comparable to the CTR+beer group (Figure 1). At the time of sacrifice, the average body weight in all groups was registered. The CTR and CTR+beer groups showed values that were not statistically significantly different (31.0 ± 1.7 and 31.9 ± 1.2 g, respectively), but those of the HFD group increased significantly (42.3 ± 4.5 g); interestingly, the coadministration of beer and HFD resulted in a significant decrease in the mean final body weight (36.6 ± 2.8 g) with respect to the HFD alone. Throughout the 75-day treatment period, the administration of beer alone had no discernible effect on any of the metabolic markers analyzed (Figure 1). This suggests that the prescribed dose did not exhibit toxicity during the specified duration.

#### 2.3.2. Assessment of Hepatic Oxidative Status

Neither the HFD nor beer treatments caused significant changes in the hepatic oxidative status of the mice compared to the controls. All groups exhibited comparable levels of both protein carbonylation and lipid peroxidation in the liver (Appendix A).

#### 2.3.3. Antisteatotic Effect of Beer

To evaluate how effective the beer treatment was in reducing hepatic steatosis, we measured the total amount of lipids in the livers of different groups of mice. As expected, mice treated for 75 days with an HFD showed significantly higher levels of liver lipids compared to both the control (CTR) and beer (CTR+beer) groups, confirming the presence of steatosis. Interestingly, mice co-treated with beer (HFD+beer) exhibited a significant reduction in steatotic conditions, as evidenced by a marked reduction in liver lipid content, comparable to the levels observed in the control and CTR+beer groups (Figure 2A). To validate the structural effects of various treatments on liver tissue, we carried out histological analysis using hematoxylin and eosin (H&E) staining. As shown in Figure 2B, liver sections from control animals displayed a healthy architecture characterized by well-defined nuclei and well-preserved cytoplasm, without any evidence of steatosis. The normal architecture was maintained in the liver tissue from the CTR+beer group, indicating that beer alone did not adversely affect liver anatomy or function. In contrast, the HFD mice showed severe changes in their liver morphology, particularly extensive macro- and microvesicular steatosis, with no inflammatory cell infiltration, likely indicating that our model represents an early stage of MASLD. Interestingly, co-treatment with beer in the HFD+beer group significantly reduced the severity of hepatic steatosis (Figure 2B). For each group, the steatosis score used to assess the severity of hepatic degeneration shown in Figure 2B is detailed in Appendix A. Consistent with the description of H&E staining provided above, a statistically significant increase was observed for the HFD group compared to the CTR, CTR+beer, and also to HFD+beer groups (Appendix A).

#### 2.3.4. Modulation of Cecal Metabolites Due to the Administration of Beer and HFD

Appendix A summarizes the concentrations (µmol/g) of various metabolites found in the cecal content of the four experimental groups. Each metabolite’s concentration is presented as the mean ± standard deviation (SD) from three replicates per animal. The analysis revealed significant metabolic differences among the groups. Lactic acid was markedly elevated in the CTR (12.80 ± 4.83) group but was significantly lower in both the HFD and HFD+beer groups. Similarly, acetic acid was highest in the CTR group (20.59 ± 5.87), with a sharp, significant decline in the HFD group. Succinic acid reached its peak concentration in the HFD+beer group (5.48 ± 0.69), followed by the CTR group (4.22 ± 1.20). Regarding metabolites represented to a lesser extent, the butyric acid concentration was found to be higher in the CTR group (0.46 ± 0.24) but decreased in both the beer and HFD groups. The formic acid concentration was highest in the CTR+beer group (1.96 ± 0.99), but showed a significant decrease in both the HFD+beer (0.11 ± 0.09) and HFD (0.17 ± 0.14) groups, with a lower concentration also observed in the CTR group (0.95 ± 1.36). Malic acid levels showed no significant differences across groups. Moreover, amino acids, including tyrosine (TYR), phenylalanine (PHE), and tryptophan (TRP), were consistently lower in the HFD groups, with TYR and PHE showing a significant reduction compared to the CTR group.

#### 2.3.5. Correlation Among Experimental Treatments and Biochemical Parameters

Pearson’s correlation coefficients were calculated to evaluate the relationships among the four different treatments to which the mice were exposed (CTR, CTR+beer, HFD, and HFD+beer) and several parameters, including blood biochemical parameters, final body weight, hepatic lipids, and metabolites in the cecal content. Interestingly, the CTR group was negatively correlated with glucose (r = −0.631), TC (r = −0.405), final body weight (r = −0.510), and hepatic lipids (r = −0.509) and positively correlated with lactic acid (r = 0.674), acetic acid (r = 0.724), propionic acid (r = 0.556), isobutyric acid (r = 0.647), butyric acid (r = 0.747), pyridoxal phosphate (r = 0.471), TYR (r = 0.642), PHE (r = 0.692), and TRP (r = 0.385). The CTR+beer group showed a negative correlation with ALT (r = −0.451), TC (r = −0.522), LDL (r = −0.519), final body weight (r = −0.372), and succinic acid (r = −0.578) and was positively correlated with formic acid (r = 0.597), isobutyric acid (r = 0.398), and pyridoxal phosphate (r = 0.400). The HFD group was negatively correlated with uric acid (r = −0.439), lactic acid (r = −0.539), acetic acid (r = −0.554), propionic acid (r = −0.539), isobutyric acid (r = −0.593), butyric acid (r = −0.453), pyridoxal phosphate (r = −0.559), TYR (r = −0.407), and PHE (r = −0.463) and positively correlated with AST (r = 0.706), ALT (r = 0.858), glucose (r = 0.649), TC (r = 0.582), LDL (r = 0.768), urea (r = 0.422), final body weight (r = 0.764), and hepatic lipids (r = 0.856). Finally, the HFD+beer group showed a negative correlation with AST (r = −0.483), lactic acid (r = −0.429), acetic acid (r = −0.466), isobutyric acid (r = −0.481), and pyridoxal phosphate (r = −0.541) and was positively correlated with succinic acid (r=0.654) (Figure 3 and Appendix A).

#### 2.3.6. Assessing Liver Transcriptomic Profile

The transcriptomic profiles of the liver tissues were obtained from four replicates (*n* = 4) for each treatment (CTR, HFD, CTR+beer, HFD+beer). About 91.5 ± 13.3 million reads were sequenced with a mapping efficiency of about 84.05% (Appendix A). In total 27,463 transcripts were identified. The Principal Component Analysis (PCA) showed a clear separation between the CTR and HFD groups (PC2 explaining the 13% of variance) but the samples seemed not to be influenced by beer consumption (Figure 4A). However, when differentially expressed genes (DEGs) were compared between the different groups, the strongest effect on the transcriptome was found for mice subjected to the HFD, with 162 DEGs for the CTRvsHFD group, and the beer consumption seemed to, in part, ameliorate the effects of the HDF, with 43 DEGs being identified for the CTRvsHFD–beer group (Appendix A). In fact, the Gene Ontology analysis showed variations in the pathways related to the response to inflammation and stilbenoids and the metabolic processing of lipids, alcohol, xenobiotic, metals, and minerals for the HFD-fed mice (Figure 4B and Appendix A), whereas for the HFD+beer group the DEGs showed variations exclusively for an acute-phase response and a response to stilbenoids.

#### 2.3.7. Assessing DNA CpG Methylation Changes in Liver

An RRBS analysis of liver tissue was obtained from four replicates (*n* = 4) for each condition (CTR, HFD, CTR+beer, HFD+beer). About 63.4 ± 17.5 million reads were sequenced with a mapping efficiency of about 76.6% and an average CpG methylation of 46.1% (Appendix A). A total of 509,561 cytosines in the CpG context with al test 10× coverage were identified in all samples. PCA showed that the liver tissues of the HFD mice showed the highest variation in CpG methylation and the biological replicates were less homogenous (Figure 5A). In order to better appreciate the methylation diversity among the different experimental conditions, PCA was performed considering only a CpG subset (*n* = 74,993), showing a significant intergroup variability (St. Dev > 10) for all comparisons tested (Figure 5B). In agreement with the RNASeq, the CTR and HFD groups exhibited the highest CpG methylation diversity. Mice clustered together for ethanol consumption, independently from diet, and were separated from the control animals for component 2 and from the HFD animals for component 1. The evaluation of CpG diversity between the mice subjected to different treatment identified 562, 429, 469, and 860 differentially methylated CpGs (FDR < 10^−6^, delta meth. > 10, at least 1 near cytosine), compared to the 230, 175, 201 and 257 genes identified for each comparison: CTRvsHFD, CTRvsHFD+beer, CTRvsCTR+beer and HFDvsHFD+beer, respectively (Appendix A). The GO analysis revealed pathways related to the acyl glycerol and lipid biosynthesis processes for CTRvsHFD+beer and pathways linked to insulin signaling for CTRvsCTR+beer comparisons (Figure 6, Appendix A). For each comparison, several genes presented a high number of long differentially methylated CpG stretches (Appendix A). Among them, the gene codifying for the lnRNA Gm26917 presented the highest number of near DMCs for the CTRvsHFD, CTRvsHFD+Beer, and CTRvsCTR+Beer comparisons, showing a long CpG rich region that was much more methylated in HFD samples (Appendix A).

#### 2.3.8. Comparison of Liver Transcriptomic and CpG Methylation Profile

The RNA-Seq and RRBS results were compared to assess possible interactions between gene methylation and expression. We observed a low correspondence between gene differential methylation and transcript abundance. Only two genes (CSAD, NAV2) for CTRvsHFD and the FGF21 gene for CTRvsCTR+beer were identified by both the RNA-Seq and RRBS (Appendix A).

## 3. Discussion

The polyphenolic composition of beer serves as a crucial quality indicator in beer processing and marketing. Indeed, the type and quantity of polyphenols determine taste, aroma, and color, as well as colloidal and foam stability, thereby impacting the shelf-life and taste of beer [25]. The quantitation results for phenolic compounds and the in vitro antioxidant activities of the beer employed for this experiment were comparable to or higher than those observed in previous studies and for some other commercial beers [26,27]. Nevertheless, since analytical methods for determining antioxidant capacity are influenced by variations in reaction mechanisms, conditions, and result interpretation, potentially confounding comparisons across studies, in vivo validation remains critical to confirm antioxidant activity under physiological conditions. In addition, the antioxidant capacity of beer is predominantly linked to phenolic compounds. Among these, phenolic acids are notable for their capability to donate hydrogen and electrons, leading to the formation of stable radical intermediates with significant antioxidant potential. Nevertheless, compounds possessing a flavonoid structure typically exhibit superior antioxidant activity compared to non-flavonoid compounds [28]. In our beer, flavonoids were by far the most abundant phenolic compounds, with a notable predominance of 3-O-quercetin derivatives as the primary flavonoid aglycones. In a study utilizing a murine model of aluminum-induced neurotoxicity, the group treated with beer exhibited significantly reduced lipid peroxidation, an increased expression of antioxidant enzymes at the mRNA level, and decreased mRNA expression of the inflammation marker TNFα [29]. The authors hypothesized that the polyphenols present in beer predominantly contribute to its antioxidative and anti-inflammatory properties observed at the brain level. These findings suggest that polyphenols derived from beer can be absorbed and enter the bloodstream to exert various biological effects. To a lesser extent, a contribution to the antioxidant potential of beer could be ascribed to the ethanol content, primarily through its ability to modulate redox reactions and enhance the bioavailability of other antioxidants. Notably, the synergy between ethanol and beer-derived polyphenols augments the overall antioxidant potential, as ethanol facilitates the solubilization and absorption of polyphenolic compounds, which exhibit robust free radical scavenging activities [30]. This interaction underscores the complex interplay of beer’s constituents in delivering protective biochemical effects. Nowadays, phenolic compounds are increasingly recognized as novel and effective approaches to alleviating or treating hepatic lipid accumulation induced by a high-fat diet and, in a broader context, MASLD [31]. Previous studies have demonstrated that a moderate daily consumption of (alcoholic) beer can counteract hepatic triglyceride accumulation in various in vivo models, such as, for example, in the study by Degrace et al. [32] using a mouse model associated with a human atherogenic lipoprotein profile (LDLr(−/−) apoB(100/100) mice). Studies have reported that isolated polyphenols, such as chlorogenic acid, can decrease the liver lipid content caused by high-fat diet consumption [33,34]. The chlorogenic acid found in green coffee bean extract was discovered to enhance fat metabolism in the livers of mice [35]. Interestingly, chlorogenic acid (88.26 ± 3.17 µg/100 mL) stands as the predominant phenolic acid found in our beer. Nevertheless, the monosaccharide flavonoid quercetin-3-O-glucoside (isoquercetin) with an average concentration of 14,684.60 µg/100 mL, is by far the most abundant phenolic compound found in our beer. Recently, various mechanisms have been proposed to explain the proven ability of isoquercetin to improve fatty liver disease in various animal models. Jin et al. [36] showed that isoquercetin regulated galectin-3-mediated insulin resistance and lipid metabolism in liver cells and provided protection against hepatic steatosis in mice with non-alcoholic steatohepatitis (NASH). In particular, the authors observed that isoquercetin can enhance insulin resistance by suppressing galectin-3. In addition, the hypolipidemic and hepatoprotective effects were linked to the restoration of expression levels of crucial genes involved in lipid metabolism to values comparable to those of the control group. Specifically, isoquercetin determined the inhibition of cholesterol synthesis and reverse transport, the modulation of fatty acid synthesis, the promotion of mitochondrial β-oxidation and the regulation of lipid metabolism [36]. In a study conducted by Qin et al. [37] the capacity of isoquercetin to ameliorate hepatic lipid accumulation in an HFD-induced MASLD rat model was attributed to its ability to activate the AMPK pathway and suppress the TGF-β pathway. In another study by Khlifi et al. [38], a plant extract particularly rich in isoquercetin and kaempferol-3-O-glucoside was shown to be capable of alleviating high-fat and fructose diet-induced fatty liver disease by modulating the metabolic and inflammatory pathways in Wistar rats. In a recent work by Zhang et al. [39], the flavonoid isoquercetin inhibited intestinal FXR-Fgf15 signaling, resulting in decreased levels of hepatic cholesterol and triglycerides. Due to its high concentration, the simple phenolic alcohol tyrosol is among the primary phenols of interest in beer, found even in alcohol-free varieties, albeit to a lesser extent. In fact, beer serves as a source of tyrosol, which originates during fermentation as a secondary metabolite of the amino acid tyrosine, facilitated by high-producing yeasts [40]. With an average concentration of 2245.90 µg/100 mL, it represents the second-most abundant phenolic compound found in our beer. In alcoholic beers, both tyrosol and its hydroxylated form hydroxytyrosol can potentially protect yeast from stress induced by high ethanol levels, similarly to what has been observed with resveratrol in wine, thus suggesting that these phenols not only undergo alterations during brewing but also influence it. Consequently, non-alcoholic beers typically exhibit a lower phenolic content, underscoring a correlation between phenols and alcohol concentrations [41]. In addition to quercetin derivatives and tyrosol, which are by far the most quantitatively representative, smaller amounts of other phenolic compounds (less than 500 µg/100 mL) were identified, including ferulic acid, gallic acid, chlorogenic acid, vanillic acid, astragalin, kaempferol 3-O-rutinoside, catechin, and epicatechin. Each of these compounds is known for having multiple beneficial properties linked to their bioactivities, and it cannot be ruled out that they may have contributed, albeit to a lesser extent, to the improvement in the pathological condition. Over the course of the 75-day treatment period, the administration of beer alone showed no noticeable impact on any of the steatosis-related indicators examined in our study. In the CTR+beer group, a slight decrease in liver transaminases was observed, with a significant negative correlation specifically with AST levels, suggesting a potential amelioration of these liver function markers. This indicated that the recommended amount was not harmful during the designated timeframe, as suggested by other previous studies. Using a higher dose of beer (0.25 mL/day, corresponding to 0.570 g of ethanol/kg b.w.), Degrace et al. [32] did not observe any signs of liver cell damage, as evidenced by the absence of ALT release into the bloodstream of LDLr(−/−) apoB100/100 mice treated for 12 weeks. In the same study, glucose, TC, LDL-C, and TG serum levels were also found to be unchanged compared to the control group, which aligns with our findings. Interestingly, in another study, although carried out in rats, the administration of 6 mL/day alcoholic (4%) beer over a 4-week period resulted in modest yet statistically significant improvements in plasma lipid and antioxidant markers, including TC, LDL-C, TG, and lipid peroxides. Notably, the effects of alcohol alone were not investigated, and the authors themselves suggested that the minimal effects observed might be attributed to the relatively low alcoholic content of the beer [42]. The findings from Jung et al. [43] and Kanuri et al. [44] demonstrated that the consumption of moderate amounts of ethanol can attenuate the progression of the early stages of MASLD in rodent models. This therapeutic effect persists when ethanol is consumed in the form of a fermented alcoholic beverage, such as beer. Additionally, these data indicate a robust association between these effects and the activation of the AdipoR1-dependent signaling cascade, which subsequently mitigates lipid peroxidation and inflammation within hepatic tissue. The dietary fiber and abundant polyphenols found in beer are known to stimulate SCFAs producing bacteria in the cecum [45]. In the present study, the metabolite concentration in the cecal content of mice showed significant variations due to different dietary conditions to which the animals were subjected. Beer supplementation led to significant changes in metabolite profiles, particularly in propionic and succinic acids. These are key cecal metabolites derived from gut microbial fermentation, playing significant roles in modulating systemic inflammation, lipid metabolism, and the gut–liver axis, which are central to MASLD pathogenesis. Propionic acid can mitigate systemic inflammation by engaging G-protein-coupled receptors (GPR41 and GPR43) on immune cells, leading to a reduced release of pro-inflammatory cytokines such as TNF-α and IL-6. This anti-inflammatory effect contributes to improved insulin sensitivity and reduced inflammatory signaling in hepatic and extra-hepatic tissues. Both succinic acid and propionic acid influence lipid metabolism. Propionic acid has been shown to suppress de novo lipogenesis by downregulating the expression of lipogenic enzymes and enhancing lipid oxidation pathways. This helps prevent hepatic fat accumulation, a hallmark of MASLD. Succinic acid, while a less-studied intermediary, has been implicated in mitochondrial function, potentially influencing lipid utilization and reactive oxygen species (ROS) generation. Through their production in the gut, these metabolites directly affect the gut–liver axis. Propionic acid reduces gut permeability, lowering endotoxin translocation into the liver, which would otherwise exacerbate inflammation and hepatic steatosis. Succinic acid, on the other hand, interacts with signaling pathways that may modulate bile acid metabolism, indirectly impacting hepatic lipid handling and inflammatory status [46,47]. These insights suggest that enhancing the gut microbial production of propionic and succinic acids, through dietary or probiotic interventions, might offer therapeutic benefits for MASLD by targeting inflammation, lipid dysregulation, and gut–liver crosstalk. These SCFAs are influenced by the microbial fermentation of polyphenol-rich compounds. Thus, the beer’s polyphenols appear to exert beneficial effects on metabolite production, particularly in a high-fat diet context. In vitro studies validate the impact of polyphenols on cecal metabolites, and animal experiments further support this interaction [48]. The decrease observed in the CTR+beer group may be attributed to the small amount of alcohol in our beer. It is reasonable to assume that this reduction did not lead to any harmful downstream effects, as we did not observe any increase in oxidative stress in either the blood or liver. This highlights, once again, the importance of both dosage and duration in the consumption of alcoholic beverages [49]. Further experiments should be performed to define the role of beer’s polyphenols in altering SCFA-producing gut microbiota. In agreement with previous studies, the differential gene expression analysis between CTR and HFD groups showed alterations in specific genes and pathways related to inflammatory response, lipid and sterol biosynthesis and the metabolism of xenobiotic compounds [17]. Within these genes, we identified different members of the serum amyloid A (SAA) and orosomucoid (ORM) family proteins. Interestingly, genes belonging to both families have been observed to be related to MASLD. SAA is related to HFD-induced obesity, and SAA1 expression can promote liver insulin resistance and intrahepatic platelet aggregation aggravating liver inflammation [50]. On the contrary, hepatic ORM2 levels markedly decreased in obese murine models and patients with MASLD and was observed also to be essential to maintain systemic lipid homeostasis [51] A similar trend was also observed in our dataset, where the HFD mice showed an increase in SAA1 and a decrease in ORM2 gene expression. We found other mis-regulated genes, such as fatty acid desaturase 2 (FADS2) and fatty acid-binding protein 5 (FABP5), whose function was related to liver lipid metabolism. FADS2 inhibition induced hepatic lipid accumulation via the impairment of very low-density lipo-protein (VLDL) secretion [52]. FABP5 played an important role in the transportation and metabolism of fatty acids in various diseases including metabolism disorders [53]. Although HFD+beer--treated animals showed alterations in genes related to inflammatory response, beer acts by restoring the expression of different genes involved in the regulation of lipid and sterol metabolism, thus explaining the improvement in MASLD observed in HFD+beer animals. As an example, beer increased the expression of the cholesterol 7 alpha-hydroxylase (CYP7A1) gene, which regulates bile acid synthesis, whose expression was downregulated in HFD mice compared to controls. A recent study aimed to evaluate the effects of brewers’ spent grain (BSG) on HFD-treated mice found that Cyp7a1 enzyme expression increased after 30% BSG supplementation, thus suggesting an increased cholesterol uptake from the blood, supported by reduced plasma total cholesterol concentrations [54]. Hepatic CYP7A1 over expression was also observed to inhibit fibroblast growth factor 21 (FGF21) expression and the ERK signaling pathway [55]. Similarly, in our model, the low expression of CYP7A1 resulted in a concurrent over expression of FGF21 in the HFD mice. FGF21 plays a major role in balancing the intake of different classes of macronutrient and, recently, FGF21 agonists were proposed as an emerging therapy for metabolic dysfunction-associated steatohepatitis [56]. Epigenetic analyses showed an influence of HFD and beer consumption on DNA methylation by separating CTR, HFD, and HFD+beer animals. Overall, methylation appeared to decrease after beer consumption, consistent with previous data reporting that ethanol intake induces global hypomethylation and alters DNA methylation [57]. It is worth noting that the observed hypomethylating capacity of beer appears to be tissue-specific, and further research could explore its potential effects on other organs, such as adipose tissue. A variation in DNA methylation affected various pathways, including those associated with acylglycerol, triglyceride, and lipid metabolic processes. Our results showed that an HFD increased DNA methylation in specific genes such as in the stearoyl-Coenzyme A desaturase 1 (SCD1) gene, which was previously observed to be over-methylated in mice fed an HFD [58]. The consumption of both beer and an HFD seemed to alter the CpG methylation in long DNA tracts, as in the case of the Gm26917 gene that codifies for a long noncoding RNA whose expression modulates the liver’s inflammatory response following acute injury [59]. Finally, the weak correspondence between DEGs and DMGs identified for each comparison indicates that transcription and the epigenetic response only partially overlapped. However, among the limited number of genes showing both gene expression and methylation variation, Cysteine Sulfinic Acid Decarboxylase CSAD and FGF21 were previously noted in two independent studies to be able to mitigate lipid accumulation and steatosis in HFD-treated mice [60,61]. HFD composition is another critical consideration in MASLD research, as its variability can introduce confounders that affect outcomes. As is commonly the case, the HFD used not only featured a high lipid content but also included significant amounts of sucrose and maltodextrin, which exacerbate lipogenesis and insulin resistance. Sugars, which are frequently added to replicate human dietary patterns, can independently drive MASLD pathogenesis, complicating the attribution of observed effects solely to dietary fat. Additionally, micronutrients, such as choline, are often limited in HFD formulations. Choline deficiency impairs hepatic lipid export via very-low-density lipoprotein secretion, amplifying steatosis irrespective of fat intake; this is avoided in our HFD, which is enriched with vitamins and minerals. Variations in fat type (e.g., saturated vs. unsaturated fats) further contribute to the differing effects on liver lipid accumulation and inflammation. Consequentially, for a rigorous study design, it is crucial to standardize and transparently report the specific HFD components to disentangle these overlapping influences and accurately interpret their relevance to MASLD development and progression. All in all, our study demonstrated that a moderate consumption of beer can prevent HFD-induced MASLD in mice. The levels of both protein carbonylation and lipid peroxidation in the liver were found to be comparable in all experimental groups, suggesting that our model represents an early stage of MASLD, typically characterized by a lack of oxidative stress. This paper suggests that the anti-steatotic effects of beer are likely attributable to its specific polyphenol content, which has been shown to restore the function of various genes involved in regulating lipid and sterol metabolism while positively influencing the metabolism of SCFAs. Nevertheless, MASLD is a multifactorial condition driven by an intricate interplay of lipid metabolism, inflammation, oxidative stress and insulin resistance. These mechanisms collectively result in hepatic steatosis and potentially progress to Metabolic dysfunction-Associated steatohepatitis (MASH) and fibrosis. As observed in the present study, moderate beer consumption influences hepatic steatosis due to its polyphenol content. Polyphenols may enhance lipid metabolism by reducing lipogenesis and promoting lipid clearance. Additionally, the bitter acids and xanthohumol from hops have demonstrated anti-inflammatory and antioxidant properties that mitigate oxidative stress and inflammation, which are crucial contributors to MASLD progression [4]. These effects counteract some aspects of MASLD pathogenesis, particularly during the early stages characterized by simple steatosis, although the evidence remains preliminary and dose dependent. The multifaceted pathogenesis of MASLD necessitates comprehensive strategies for prevention and management, targeting lipid regulation, inflammation, and insulin sensitivity. While a moderate consumption of beer may confer benefits due to the presence of bioactive compounds, excessive alcohol consumption is a risk factor for liver disease. Further clinical research is required to validate the potential protective effects of beer-derived polyphenols in MASLD contexts. Accordingly, despite growing research interest in this field and the potential benefits identified, the authors of this article strongly advise against alcohol consumption by at-risk groups. In particular, for groups like children, pregnant women, individuals with liver diseases, or those on medication, no safe amount of alcohol consumption has been established and our recommendation is to abstain entirely from alcohol. Furthermore, for healthy people, alcohol consumption should consistently accompany meals, and excessive intake should be strictly avoided. Among the limitations of this study is the reliance on a single animal model; future research should include other MASLD models or human trials to validate these findings. Moreover, our findings are specific to the beer type tested and refer to a single dose; accordingly, it would be interesting to explore effects across diverse beer formulations and also varying doses to identify thresholds for toxicity and efficacy.

## 4. Materials and Methods

### 4.1. Beer Sample

The sample bottles of beer required for experimental purposes were purchased from Birrificio Centolitri (Baschi, Terni, Italy). The beer was an unpasteurized ale refermented in the bottle, with a 4.5% (*v*/*v*) alcohol content. The cereals used for beer production were primarily barley (95%), but also rye and wheat (remaining 5%), all in a malted form. It is a single-hop beer featuring hops originally sourced from the Czech Republic but cultivated in Italy. The yeast used was Belgian Fermentis T58 (Lesaffre, Marcq-en-Barœul, France). Beer samples were stored at 4 °C in a refrigerator until use.

### 4.2. Antioxidant Profiling of Beer

#### 4.2.1. Bioactive Molecules Content

The evaluation of the overall phenolic content was carried out utilizing the Folin–Ciocalteu colorimetric method as described by Singleton et al. [62]. The results are expressed as milligrams of gallic acid equivalent per 100 mL (mg GAE/100 mL). The determination of total flavonoid content was performed by employing the aluminum chloride colorimetric method introduced by Kim et al. [63], and the outcomes are presented in milligrams of catechin equivalent per 100 mL (mg CE/100 mL). The quantification of total flavonols, expressed as milligrams of quercetin equivalent per 100 mL (mg QE/100 mL), followed the procedure outlined by Romani et al. [64].

#### 4.2.2. Phenolic Compounds Profiling by UHPLC-ESI-MS/MS Analysis

Key phenolic compounds were systematically chosen for an exhaustive quantitative analysis of the extracts employing a Sciex 5500 QTrap+ mass spectrometer (AB Sciex LLC, Framingham, MA, USA) coupled with Ultra-High-Performance Liquid Chromatography using Electrospray Ionization Tandem Mass Spectrometry (UHPLC-ESI-MS/MS). The mass spectrometer was outfitted with a Turbo V ion-spray source and interfaced with an ExionLC AC System, specifically crafted by Shimadzu (Shimadzu Corporation, Kyoto, Japan). This system comprised two ExionLC AC pumps, an autosampler, a controller, a degasser, and a tray. MS/MS experiments were conducted in the electrospray negative-ion mode, employing nitrogen as the collision gas. Operational source parameters encompassed turbospray as the source type, nebulizer gas (GS1) set at 70, turbo gas (GS2) at 50, curtain gas (CUR) at 10, temperature (TEM) at 500 °C, Ionspray Voltage (IS) at −4500 V, and entrance potential (EP) at 10 V. Compound parameters, including declustering potential (DP), collision energy (CE), and collision cell exit potential (CXP), were meticulously adjusted for the specific Selected Reaction Monitoring (SRM) transition of each component. The analyses were conducted in triplicate, and the outcomes are presented in micrograms per 100 mL of dry weight.

#### 4.2.3. Assessment of Antioxidant Activity Through In Vitro Assays

The in vitro antioxidant potential of beer was evaluated through a comprehensive methodology that integrated fluorimetric and spectrophotometric techniques. The assessment of 2,2-diphenyl-1-picrylhydrazyl (DPPH) radical scavenging activity followed the protocol established by Sokmen et al. [65]. The determination of Oxygen Radical Absorbance Capacity (ORAC) for beer adhered to the procedures outlined by Bacchiocca et al. [66]. To quantify the antioxidant capacity of beer, the Ferric Reducing Antioxidant Power (FRAP) assay, as detailed by Colosimo et al. [67], was employed. The results were expressed as milligrams of Trolox equivalent per 100 mL (mg TE/100 mL).

### 4.3. In Vivo Experiment: Hepatoprotective Assay

#### 4.3.1. Animal Procedure

The in vivo experiment was performed using forty-eight male C57BL/6J mice of about 25 g body weight (b.w.). The animals were divided into four groups, housed in cages subjected to a 12 h light and dark cycle at room temperature with a relative humidity of 55%, and provided with unrestricted access to drinking water and food. The four groups, of twelve animals each, were divided as follows: (1) control mice (CTR), (2) mice supplemented daily fpr 75 days with beer at the dose of 0.14 mL/day in drinking water (corresponding to 0.132 g/kg b.w. of ethanol per day) (CTR+beer); (3) mice fed with a high-fat diet for 75 days (HFD); (4) mice fed with a high-fat diet and supplemented daily with beer at the dose of 0.14 mL/day (corresponding to 0.132 g/kg b.w. of ethanol per day) for 75 days (HFD+beer). The beer dose administered to the mice (0.14 mL/day) was calculated based on a daily intake corresponding to a 70 kg human consuming 400 mL of beer per day. This value is equivalent to 15.8 g ethanol/day and falls within the tolerability range for minimal risk (14 to 28 g ethanol per day), as indicated by the Dietary Guidelines for Americans, 2020–2025 [68]. A standard feed was administered to the CTR and CTR+beer groups in a pellet, containing 19.0% proteins, 6.0% fibers, 7.0% minerals and vitamins moisture, 64.0% carbohydrates, and 4.0% fats. An HFD was administered to the HFD and HFD+beer groups in a pellet, containing 24.4% proteins, 6.0% fibers, 5.3% minerals and vitamins moisture, 9.4 sugars, and 34.6% fats. Table 3 shows the detailed composition of the HFD. The weight of each animal was recorded weekly from the beginning of the experiment until the end of the experiment when the animals were euthanized. Prior to euthanasia, blood samples were obtained from each animal in the four experimental groups via cardiac puncture under general anesthesia. These samples were then centrifuged at 4000 rpm for 15 min to obtain plasma for subsequent laboratory analysis. The animals were euthanized by removing their hearts after blood collection. Livers were weighed and liver samples were stored at −80 °C for lipid extraction and quantification, evaluation of biochemical markers of oxidative stress, and for total DNA and RNA extraction, or preserved in a solution consisting of 70% ethyl alcohol and 30% distilled water [15,69] at 4 °C for histopathological analysis. The research protocol received approval from the Local Ethical Committee, in accordance with Italian regulations on the ethical treatment and utilization of animals for scientific research (Legislative Decree 26/2014) and the European Union Directive 2010/63/EU concerning animal experimentation. Additionally, the experiment’s protocol was formally authorized by the relevant commission of the Italian Ministry of Health (ministerial approval n. 873/2021-PR).

#### 4.3.2. Analysis of Biochemical Parameters

The levels of total cholesterol (TC), low-density lipoprotein cholesterol (LDL-C), triglycerides (TG), urea, glucose, and the enzyme activities of ALT and AST were assessed following the guidelines provided by the manufacturer, using commercial assays conducted at a specialized laboratory (PAIMBiolabor, Livorno, Italy).

#### 4.3.3. Biomarkers of Oxidative Stress in the Liver

The concentration of malondialdehyde (MDA) in the liver samples was evaluated using the method outlined by Seljeskog and colleagues [70], with minor adjustments as specified in our prior publication [71]. MDA concentration was measured in nanomoles per gram of tissue (nmol MDA/g tissue). Protein oxidation levels were determined using the carbonyl protein assay based on the protocol by Terevinto et al. [72], with slight modifications as detailed in Pozzo et al. [71]. Concentrations of carbonylated proteins were calculated in nanomoles per gram of tissue (nmol protein carbonyls/g tissue).

#### 4.3.4. Quantitation of Liver Lipids to Estimate MASLD

The amount of lipids in the liver was measured using the gravimetric method developed by Folch et al. [73], with slight modifications. Rat liver samples were mixed with equal amounts of water and methanol before being homogenized. This mixture underwent three successive extractions with chloroform, followed by two washes with 1 M KCl and water. After complete evaporation and prolonged drying of the chloroform solution (until a constant weight was reached), the lipid content was measured and expressed as mg/g of tissue (mg/g tissue).

#### 4.3.5. Histopathological Analysis

Following the sacrifices of experimental animals, necropsies were conducted in accordance with laboratory standard operating procedures (SOPs). Livers were collected and preserved in a solution consisting of 70% alcohol (a combination of ethyl and isopropyl alcohol in proportions of approximately 60% and 40%, respectively) and 30% distilled water. Trimming procedures were carried out as per SOPs. Each trimmed liver specimen underwent processing and embedding in paraffin blocks following laboratory SOPs. Subsequently, 5 µm sections were sliced and routinely stained with hematoxylin–eosin (HE). Histopathological evaluation was systematically performed in a blinded manner, without prior knowledge of the experimental groups, under a light microscope [65]. The severity of hepatic degeneration was assessed using a semi-quantitative scoring system, defined as follows: a score of 0 indicates <5% of hepatocytes are affected by lipid vacuoles; 0.5 indicates involvement of 5–15% of hepatocytes; 1 corresponds to 15–30% involvement; 2 indicates 30–60% of hepatocytes are affected; 3 represents 60–80% involvement; and 4 signifies >80% of hepatocytes exhibit lipid vacuolation. This scoring methodology was proposed by the NASH Clinical Research Network [74], slightly modified [75].

#### 4.3.6. Evaluation of Cecal Metabolomic Changes

For sample preparation, approximately 45 mg of the cecal content samples were thawed and dissolved in 500 µL of bidistilled H_2_O (ELGA Ultrapure Laboratory Water). The sample was homogenized by vortexing and sonication (40 kHz for 5 min). After sonication, the samples were centrifuged at 14,000 rpm for 10 min. The supernatant was deproteinized by ultracentrifugation (30 min) using Microcon^®^ Centrifugal Filters with a cut-off of 3 kDa (Merk, Milan, Italy) (named 3 kDa filtered samples). The 3 kDa filtered samples were 3-fold diluted in 5 mM sulfuric acid, filtered using a 0.20 μm RC Mini-Uniprep (Agilent Technologies, Milan, Italy) filter, injected in the HPLC system equipped with diode array detector (DAD) and fluorescence detector (FD) (Vinj = 5 μL), and analyzed as previously reported. An Agilent 1260 Infinity HPLC system (Agilent Technologies, Santa Clara, CA, USA) (G1311B quaternary pump) equipped with a 1260 Infinity High Performance Degasser, a TCC G1316A thermostat, a 1260ALS autosampler (G1329B), and a UV/vis diode array (1260 DAD G4212B) was employed. The identification of SCFAs was based on the comparison of the retention time and the UV spectra of standard compounds. The 220 nm detection was selected to control the interference of high absorbing compounds. The chromatographic separation was carried out by Zorbax Phenyl-Hexyl RP C18 (Agilent Technologies, Santa Clara, CA, USA) 250 × 4.6 mm (silica particle size 4 μm) at 45 °C using the following elution profile: 15 min isocratic elution with 0.1% phosphoric acid (pH 2.2), followed by a 10 min gradient to 80% methanol, and 10 min isocratic elution in 80% methanol (flow 0.8 mL/min). The column was rinsed with 100% methanol for 15 min and a re-equilibration step was performed. All the solutions were filtered using a 0.22 μm regenerate cellulose filter (Millipore, Milan, Italy) [76].

### 4.4. Analysis of Gene Expression and DNA Methylation

#### 4.4.1. Nucleic Acids Isolation

Total RNA was extracted from liver tissue with Trizol, following the manufacturers’ instructions. The upper aqueous phase solution containing RNA was purified with NucleoSpin miRNA kit (Macherey-Nagel, Düren, Germany), following the protocol in combination with TRIzol (Invitrogen, Carlsbad, CA, USA) lysis with small and large RNA in one fraction (total RNA). RNA concentration and quality were determined by Agilent 2100 with RNA 6000 Nano Kit (Santa Clara, CA, USA). Genomic DNA from liver tissue was isolated using the NucleoSpin Tissue kit (Macherey-Nagel, Düren, Germany), following the manufacturer’s instructions. DNA concentration was calculated using nanodrop and DNA integrity assessed by gel electrophoresis. The isolated RNAs and DNAs were stored at −80 °C and −20 °C until use.

#### 4.4.2. Library Preparation and Sequencing

Whole-transcriptome sequencing libraries were being outsourced using the Watchmaker RNA with Polaris Depletion kit. Reduced Representation Bisulfite Sequencing libraries were generated after MspI digestion using the TruSeq DNA PCR-Free Library Preparation Kit (Illumina, San Diego, CA, USA). Bisulfite treatment was performed after ligation of adapters using the EpiTect Bisulfite Kits (Qiagen, Venlo, The Netherlands), and finally, the libraries were enriched with KAPA HiFi Uracil (Kapa Biosystems, Wilmington, MA, USA). Both RNA-Seq and RRBS libraries were sequenced on a NovaSeq X instrument (Illumina, San Diego, CA, USA) to generate 150-base paired-end reads.

#### 4.4.3. Bioinformatic Data Analysis

Preliminary quality control of RNA-Seq and RRBS raw reads was carried out with FastQC v0.11.9 (http://www.bioinformatics.babraham.ac.uk/projects/fastqc/, accessed on 11 December 2024). RNA-Seq raw sequences were run with the nf-core/rnaseq version 3.8.1 pipeline (https://nf-co.re/rnaseq, accessed on 11 December 2024). The pipeline integrates TrimGalore version 0.6.7 and STAR version 2.7.10a [77] for sequence trimming and alignment. Sequences were aligned to the mouse GRCm39 reference genome. Salmon version 1.5.2 was used to quantify alignments to gene regions (https://combine-lab.github.io/salmon/, accessed on 11 December 2024). The EdgeR Bioconductor package version 3.6 was used to estimate differential expression between different comparisons (Bioconductor, https://bioconductor.org/packages/release/bioc/html/edgeR.html, accessed on 11 December 2024). RRBS raw sequences were filtered with TrimGalore v0.6.4_dev (http://www.bioinformatics.babraham.ac.uk/projects/trim_galore/ accessed on 11 December 2024) to remove low-quality bases and adapters, using RRBS-specific parameters. The Bismark software v.0.22.3 (https://www.bioinformatics.babraham.ac.uk/projects/bismark/ accessed on 11 December 2024) was used to align each read to a bisulfite-converted Mus musculus reference genome (GRCm39: GCF_000001635.27), and the Bismark methylation_extractor function was used to extract methylation calls. For visualization and analysis of the Bismark output of the Seqmonk software (version 1.48.0) was used (http://www.bioinformatics.babraham.ac.uk/projects/seqmonk/ accessed on 16 December 2024). Only positions with a depth of at least 10 cytosines were recorded in all samples and used for RRBS analysis. Differentially methylated cytosines (DMCs) between all comparisons were obtained by using the logistic regression filter in R (FDR < 10^−6^, differential methylation percentage ≥ 10, at least 1 near cytosine). Visualization of CpG methylation level was performed using the Methylation plotter Software (http://maplab.imppc.org/methylation_plotter/ accessed on 16 March 2024). Gene ontology (GO). classification was performed on DEGs and genes close to DMCs (≤2000 bp distance), using the Cytoscape plug-in ClueGO, which integrates GO and enhances biological interpretation of large lists of genes.

### 4.5. Statistical Analysis

The findings are depicted as the mean value (*n* = 3) ± standard deviation (SD). The transcriptomic profile of liver tissue was obtained from four replicates (*n* = 4). In the in vivo study, significant differences among the means of the four mice groups were analyzed using a one-way analysis of variance (ANOVA) followed by Tukey’s post hoc test, with significance at *p* ≤ 0.05; the exception was metabolites concentration, that was analyzed by two-way ANOVA. All analyses were carried out using Prism, GraphPad Software, version 8.0.1, based in San Diego, CA, USA. A Pearson’s correlation analysis was conducted to evaluate the correlation between the four experimental groups (CTR, CTR+beer, HFD and HFD+beer) and some variables (blood parameters, body weight, hepatic lipids and cecal metabolites variables) using XLSTAT software (version 2019).

## Figures and Tables

**Figure 1 molecules-29-05954-f001:**
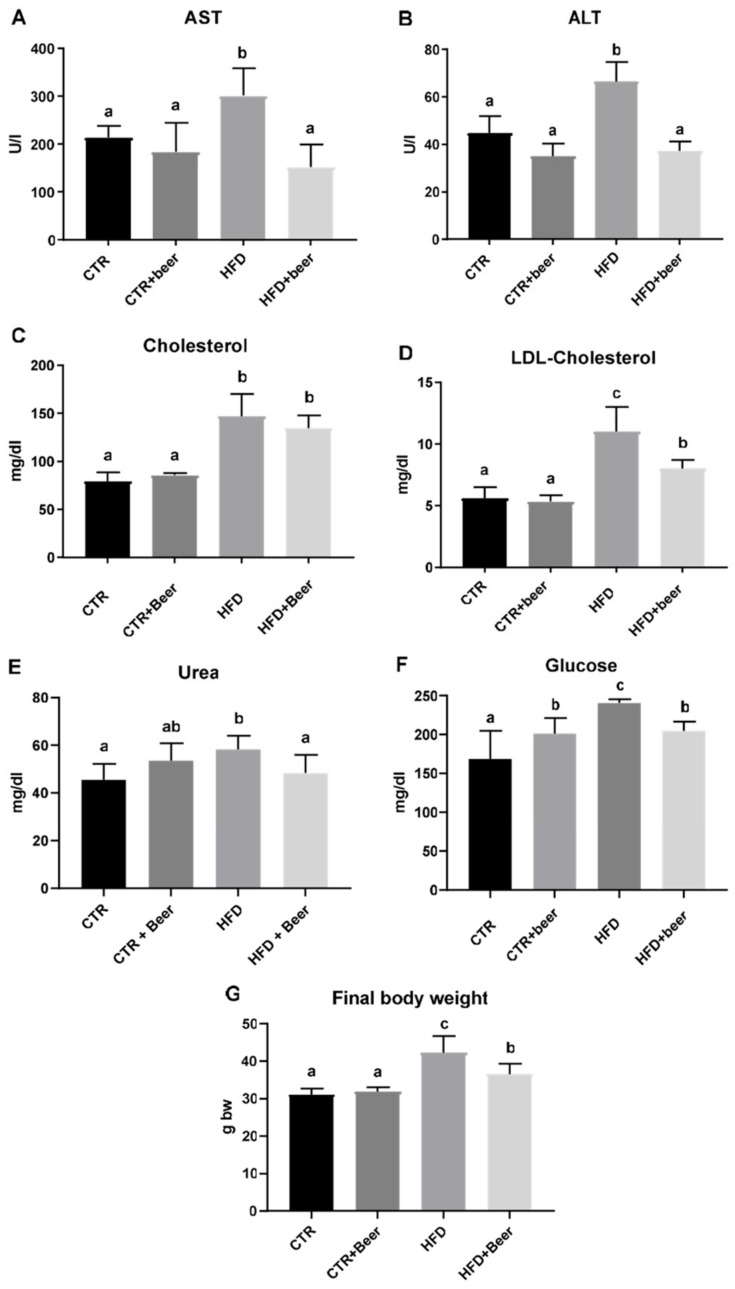
Biochemical parameters in the mouse serum (aspartate aminotransferase (AST) (**A**), alanine aminotransferase (ALT) (**B**), cholesterol (**C**), low-density lipoprotein (LDL)-cholesterol (**D**), urea (**E**), glucose (**F**)) and body weight at sacrifices (**G**) of CTR, CTR+beer, HFD and HFD+beer mice. Results are reported as means ± SD of three replicates. Values within each row with different letters (a, b, c) are significantly different according to one-way ANOVA (*p* ≤ 0.05) followed by Tukey’s post hoc test.

**Figure 2 molecules-29-05954-f002:**
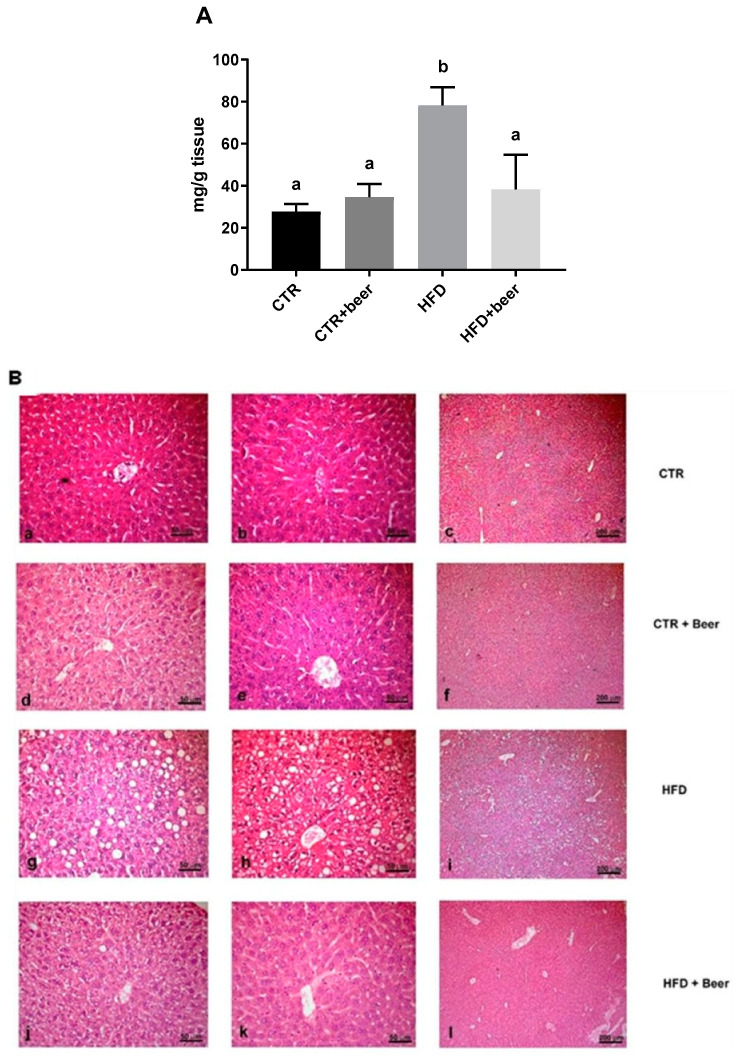
Total hepatic lipid content (**A**) measured in liver tissue from all CTR, CTR+beer, HFD and HFD + beer mice. Values are expressed as means ± SD. a, b: Values significantly different according to one way ANOVA-test (*p* ≤ 0.05) followed by Tukey’s post hoc test. Hematoxylin and eosin (H&E) staining (**B**) of liver tissue from CTR, CTR+beer, HFD and HFD+beer mice. Magnification: (**a**,**b**,**d**,**e**,**g**,**h**,**j**,**k**), bar = 50 µm; (**c**,**f**,**i**,**l**), bar = 200 µm.

**Figure 3 molecules-29-05954-f003:**
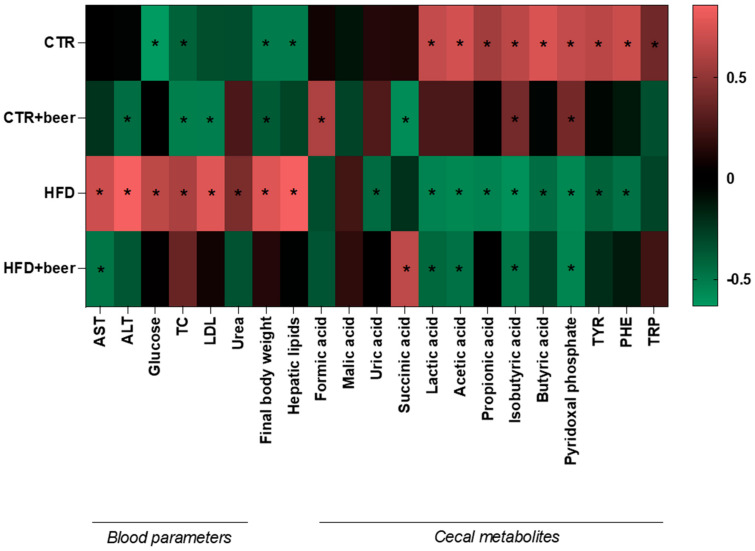
Heat map reflecting Pearson’s correlation coefficients between experimental treatments (*n* = 8) and some parameters analyzed during the study, specifically blood parameters, final body weight, hepatic lipids and cecal metabolites. Red color represents a positive correlation, and green color represents a negative correlation. * Represents a significant positive or negative correlation.

**Figure 4 molecules-29-05954-f004:**
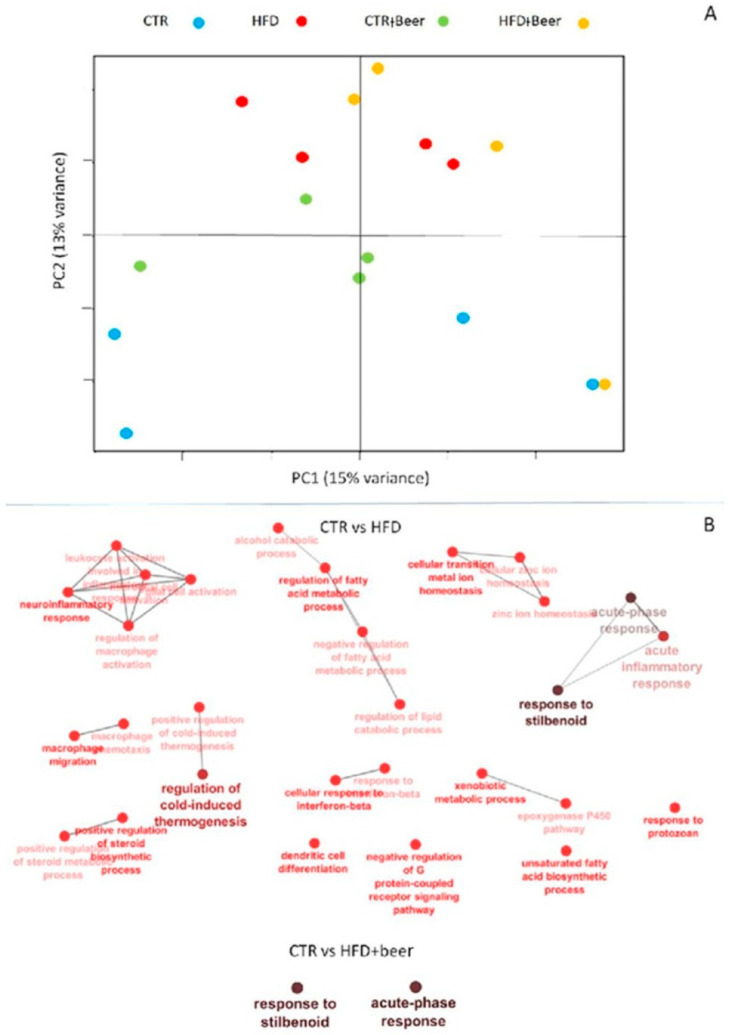
(**A**) Principal Component Analysis (PCA) of 20,117 transcripts with at least 1 count for 100,000 reads in at least 3 samples. The groups are identified as follows: CTR in blue, CTR+beer in green, HFD in red, HFD+beer in yellow. (**B**) ClueGO Gene Ontology GO analysis to compare (**A**) 162 DEGs for CTR vs. HFD and (**B**) 43 DEGs for CTR vs. HFD+beer.

**Figure 5 molecules-29-05954-f005:**
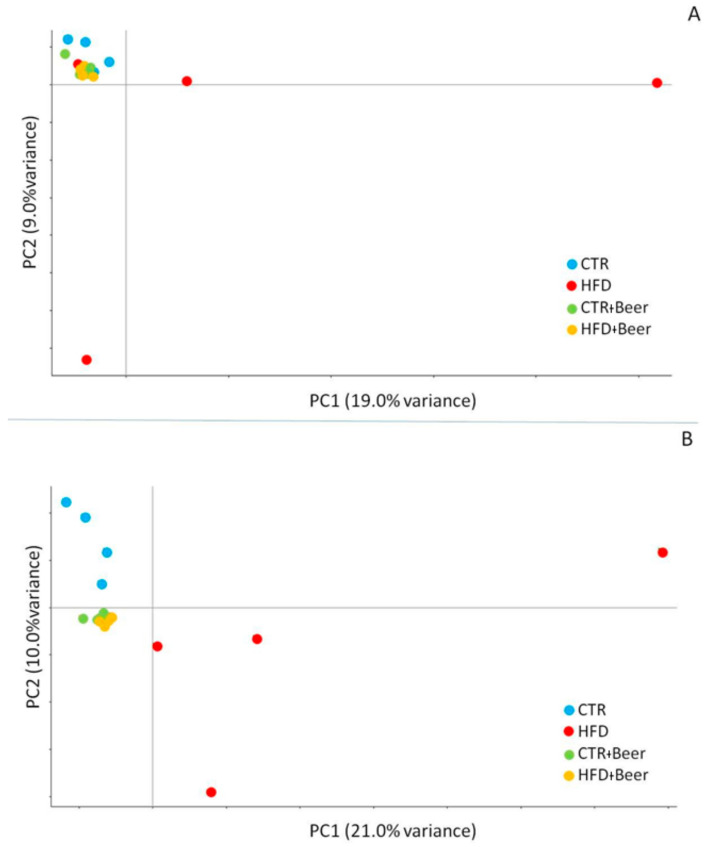
Principal Component Analysis (PCA) of CpG methylation level for (**A**) 509,561 cytosines in CpG context with al test 10X coverage were identified in all samples and (**B**) a subset of 74,993 cytosines in the CpG context with highest inter-group variability. The groups are identified as follows: CTR in blue, CTR+beer in green, HFD in red, HFD+beer in yellow.

**Figure 6 molecules-29-05954-f006:**
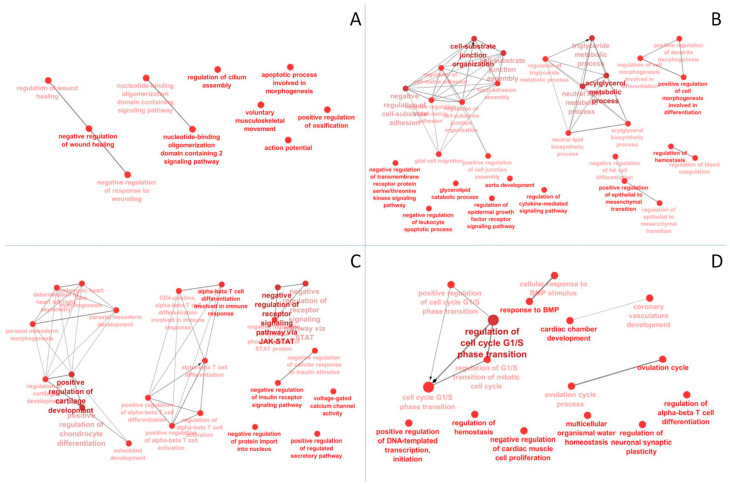
ClueGO Gene Ontology GO analysis for comparisons between (**A**) 230 DMGs for CTRvsHFD and (**B**) 175 DMGs for CTRvsHFD+beer, (**C**) 201 DMGs for CTRvsCTR+beer, (**D**) and 257 DMGs for HFDvsHFD+beer.

**Table 1 molecules-29-05954-t001:** Bioactive compounds and in vitro antioxidant activity of beer.

		*Beer*
Bioactive compounds	Total polyphenols (mg GAE/100 mL)	25.01 ± 1.27
	Flavonoids (mg CE/100 mL)	3.17 ± 0.17
	Flavonols (mg QE/100 mL)	3.07 ± 0.23
Antioxidant activity	ORAC (mg TE/100 mL)	103.10 ± 7.01
	FRAP (mg TE/100 mL)	39.11 ± 0.42
	DPPH (mg TE/100 mL)	154.77 ± 13.05

Values are reported as means of three replicates ± s.d.

**Table 2 molecules-29-05954-t002:** Content of individual phenolic compounds in beer (µg/100 mL).

Compound Name	µg/100 mL
Gallic Acid	62.70 ± 2.00
3-*O*-Caffeoylquinic acid (Chlorogenic acid)	88.26 ± 3.17
Protocatechuic acid	14.68 ± 1.99
Caffeic Acid	11.91 ± 1.40
Vanillic Acid	69.29 ± 3.20
p-Coumaric Acid	2.38 ± 0.20
*trans*-Ferulic Acid	29.87 ± 2.28
Rosmarinic Acid	0.09 ± 0.03
**∑ Phenolic acids**	**279.17**
Quercetin	0.54 ± 0.24
Quercetin 3-O-glucoside	14,684.60 ± 2678.60
Quercetin 3,4-O-diglucoside	10.56 ± 1.35
Quercetin 3-O-rutinoside (Rutin)	169.43 ± 30.25
Kaempferol 3-*O*-glucoside (Astragalin)	99.59 ± 10.30
Kaempferol 3-*O*-rutinoside	87.99 ± 3.90
**∑ Flavonols**	**15,052.71**
(+)-Catechin	415.29 ± 37.40
(−)-Epicatechin	85.88 ± 2.40
Apigenin	0.09 ± 0.03
**∑ Flavan-3-ols**	**501.27**
Tyrosol	2245.90 ± 56.00
Hydroxytyrosol	8.93 ± 0.41
Naringenin	3.56 ± 0.45
Erodictyol	0.73 ± 0.25
Luteolin	0.51 ± 0.20
Phloridzin	0.28 ± 0.05
Resveratrol	0.19 ± 0.09
Oleuropein	0.14 ± 0.11
Verbascoside	0.08 ± 0.05
Phloretin	0.02 ± 0.00
**∑ Others**	**2259.84**

Values are reported as means of three replicates ± s.d.

**Table 3 molecules-29-05954-t003:** Composition of rodent diet with 60 Kcal % fat, D12492, Diets INC (HFD).

Class Description	Ingredient	Grams
Protein	Casein, Lactic, 30 Mesh	200.00
Protein	Cystine, L	3.00
Carbohydrate	Lodex 10	125.00
Carbohydrate	Sucrose, Fine Granulated	72.80
Fiber	Solka Floc, FCC200	50.00
Fat	Lard	245.00
Fat	Soybean Oil, USP	25.00
Mineral	S10026B	50.00
Vitamin	Choline Bitartrate	2.00
Vitamin	V10001C	1.00
Dye	Dye, Blue FD&C #1, Alum. Lake 35–42%	0.05
	Total:	773.85

## Data Availability

All sequence data are deposited at the NCBI Sequence Read Archive (SRA) (https://www.ncbi.nlm.nih.gov/sra) (Accession Number PRJNA1088980).

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
