# Peer review of "A Moderate Intake of Beer Improves Metabolic Dysfunction-Associated Steatotic Liver Disease (MASLD) in a High-Fat Diet (HFD)-Induced Mouse Model"

_molecules, 2024, doi:10.3390/molecules29245954_

Round 1
Reviewer 1 Report
Comments and Suggestions for Authors
This is an interesting study, but several points need clarification.
MAJOR COMMENTS:
1. The manuscript attributes the antioxidant properties of beer entirely to polyphenols and bitter acids without considering other factors like ethanol's minor contribution.
2. Why was the C57BL/6J mouse strain chosen for this study, and how well does it represent human NAFLD pathophysiology?
3. Please include a brief discussion on ethanol's potential role as an antioxidant in small amounts, emphasizing its synergistic effects with polyphenols.
4. The definition of "moderate" alcohol consumption is vague and inconsistently referenced across the text. Please clearly define "moderate" in terms of grams of ethanol per day and link it to global health guidelines (e.g., WHO recommendations).
5. The study claims generalizability of findings using one HFD-induced NAFLD model. Please acknowledge the limitations of using a single animal model and suggest validation in other NAFLD models or human trials for future study.
6. The manuscript states that no oxidative stress was observed but does not explore whether the early stage of NAFLD inherently prevents oxidative stress. Please discuss whether the absence of oxidative stress markers is due to the disease stage or the intervention itself.
7. The RNA-seq findings lack a detailed explanation of why certain DEGs (differentially expressed genes) are relevant to NAFLD progression. Please elaborate on specific DEGs linked to NAFLD and how they connect to the observed biochemical changes.
8. The paper claims CpG methylation changes "ameliorate NAFLD" but fails to establish a direct cause-effect relationship. Please clarify the indirect nature of CpG methylation's effects and suggest further studies to confirm causation.
9. The link between beer consumption and SCFA metabolism is asserted without showing causal pathways. Propose experiments or previous study to directly measure the role of beer's polyphenols in altering SCFA-producing gut microbiota.
10. Histological analysis describes steatosis reduction but does not quantify the severity or provide robust grading criteria. Please introduce standardized steatosis grading (e.g., NAS or SAF scores) for better interpretation.
11. Results are generalized as representative of "beer," ignoring variability in beer types and production processes. Please emphasize that findings are specific to the tested beer type and suggest exploring effects across diverse beer formulations.
12. How were the beer samples stored and handled before administration to ensure consistency in polyphenol content throughout the experiment?
13. The manuscript focuses on quercetin and isoquercetin but neglects the contributions of other phenolic compounds like ferulic acid. Briefly compare the relative abundance and bioactivity of all detected phenolic compounds.
14. The claim that transcriptomic changes in NAFLD are reversible lacks sufficient support. Please cite more robust evidence or acknowledge that reversibility may depend on the treatment duration or disease stage.
15. The manuscript attributes high ORAC, FRAP, and DPPH values solely to phenolic compounds without considering methodological biases. Please discuss potential confounders in these assays and highlight the need for in vivo validation of antioxidant activity.
16. The study uses one beer dose without exploring dose-response effects. Please acknowledge this limitation and suggest further studies with varying doses to identify thresholds for toxicity and efficacy.
17. The link between beer and NAFLD pathogenesis is oversimplified, ignoring complex interactions of lipids, inflammation, and insulin resistance. Please expand on the multifactorial nature of NAFLD and situate beer’s role within these mechanisms.
18. The significance of changes in metabolites like succinic acid and propionic acid is inadequately explained. Please elaborate on how these metabolites impact systemic inflammation, lipid metabolism, or gut-liver axis.
19. The CTR+beer group is underexplored in comparison to CTR, HFD, and HFD+beer groups. Discuss any unique insights from the CTR+beer group, such as baseline metabolic changes or adaptive responses.
20. The manuscript does not detail the HFD composition's potential confounders, like sugar content or micronutrients. Provide a comprehensive breakdown of the HFD components and their relevance to NAFLD.
21. The claim that beer reduces global methylation does not account for tissue-specific effects. Please state that findings are liver-specific and propose exploring effects in other organs like adipose tissue.
22. The manuscript prematurely extrapolates findings to human health without addressing translational challenges. Please emphasize the preliminary nature of findings and propose human studies to validate results.
23. The manuscript advises against excessive alcohol consumption but does not address safe limits for at-risk groups. Please provide explicit recommendations for groups like pregnant women, individuals with liver diseases, or those on medication.
MINOR COMMENTS:
1. Line 22-23: "Beer and its components have shown potential in reducing hepatic steatosis in various rodent models." Revision suggestion: Rephrase as "Beer and its components show potential for reducing hepatic steatosis in rodent models."
2. Line 52: "There is a poorness of epidemiological studies..." Revision suggestion: Replace "poorness" with "lack."
3. Line 92: "The beer displayed a total polyphenols content..." Revision suggestion: Correct to "displayed total polyphenol content."
4. Line 152: "Neither the HFD nor the beer treatment caused significant changes in..." Revision suggestion: Streamline as "Neither the HFD nor beer treatments caused significant changes..."
5. Line 177-179: "The analysis reveals distinct metabolic variations among groups." Revision suggestion: Clarify as "The analysis revealed significant metabolic differences among the groups."
6. Line 305: "Studies have reported that individual isolated polyphenols..." Revision suggestion: Remove "individual" to avoid redundancy.
7. Line 421-423: "Authors strongly advise against alcohol consumption among..." Revision: suggestion: Add specificity: "The authors strongly advise against alcohol consumption by children..."
OTHERS:
Consistency
1. "nonalcoholic fatty liver disease" should be consistently written as either "non-alcoholic fatty liver disease (NAFLD)" or "nonalcoholic fatty liver disease" for consistency.
2. "hepatic inflammatory response to steatosis along with the activation": consider rephrasing for clarity as "hepatic inflammatory response transitioning to steatosis, alongside the activation."
3. "high fat diet": should be hyphenated as "high-fat diet."
Stylistic Adjustments:
4. "Drinking beer in low to moderate amounts": revise for clarity as "Consuming beer in low to moderate amounts."
5. "but it’s thought to be related to": replace contraction "it’s" with "it is."
6. "the prescribed dose was non-toxic": revise to "the prescribed dose did not exhibit toxicity."
7. "suggesting our model represent an early stage of NAFLD": revise to "suggesting that our model represents an early stage."
8. "The type and quantity of polyphenols influence taste, aroma, color": consider "determine" instead of "influence."
9. "was observed to modulate liver inflammatory response after acute liver injury": revise as "modulated the liver’s inflammatory response following acute injury."
10. "interesting variations in metabolite profiles": suggest replace "interesting" with "notable."
11. "short chain fatty acids (SCFA) metabolism": revise to "short-chain fatty acid metabolism (SCFA)."
Sentence Structure and Clarity:
12. "The results obtained for the quantitation of phenolic compounds". consider revise with: "The quantitation results for phenolic compounds."
13. "was subjected to three successive extractions with chloroform": clarify with "underwent three successive extractions."
14. "variation in the DNA methylome status of genes": did the author means "variations in the DNA methylation status."?
15. "molecular biology characterization of liver alteration in HFD mouse model provided deeper understanding". revision suggestion: "The molecular characterization of liver alterations in HFD mouse models provided deeper insights."
Author Response
Responses to Reviewer #1:
Thank you very much for taking the time to review this manuscript. Please find the detailed responses below and the corresponding revisions/corrections highlighted/in track changes in the re-submitted files.
Major comments
1. Ethanol's minor contribution to potential antioxidant activity of beer has been added in the discussion with reference. Reference numbers have been modified accordingly.
2. C57BL/6J mouse strain was chosen for this study because it well represents human NAFLD pathophysiology. In literature, numerous mouse models of NAFLD have been developed, including various dietary approaches, and the majority of such studies utilize the C57BL/6 mouse strain, due to their availability and low cost, detailed phenotypic characterization, and overall favoured standing as the background strain for most genetically engineered models (Karimkhanloo, H., Keenan, S. N., Bayliss, J., De Nardo, W., Miotto, P. M., Devereux, C. J., ... & Montgomery, M. K. (2023). Mouse strain-dependent variation in metabolic associated fatty liver disease (MAFLD): a comprehensive resource tool for pre-clinical studies. Scientific reports, 13(1), 4711).
3. Connected to point 1), we have included in the text a brief discussion regarding ethanol as an antioxidant in small amounts, emphasizing its synergistic effects with polyphenols.
4. The beer dose administered to the mice (0.14 mL/day) was calculated based on a daily intake corresponding to a 70 kg human consuming 400 mL of beer per day (equivalent to 15.8 g ethanol/day). Most of Western countries set daily limits range for a low risk due to alcohol intake from 10-48 g per day for both men and women. In US, for example, for a man the limit ranges from 14 to 28 g (1-2 US standard drinks) (Snetselaar et al., Dietary Guidelines for Americans, 2020-2025). The conversion from human to mouse dosage was performed using the established formula described in the literature (Reagan-Shaw et al., 2007. Dose translation from animal to human studies revisited. FASEB J. 22, 659–661). The reference dose of alcoholic beer for humans is well below the safety threshold, as previously reported (Daimiel et al., 2021). This has been added in the text in the M&M paragraph 4.3.1.
5. Limitations of using a single animal model and the need of validation in other NAFLD models or human trials for future study have been added in the text (end of Discussion paragraph).
6. The absence of oxidative stress markers is due to the disease stage. Oxidative stress is often considered part of the “second hit” for the progression of the disease to NASH. It has been better explicated in the discussion.
7. In the discussion section, we add example of genes potentially related to NAFLD, such as FABP5, FADS2, CYP7A1 and FDF21.
8. For the CpG methylation, in agreement with previous studies we only observed a global hypomethylation induced by ethanol consumption. In addiction we observed specific CpG methylation changes for all the comparison tested. We did not claim that methylation changes "ameliorate NAFLD". The ameliorating effect due to low consumption of beer was observed only for transcriptomic data.
9. The following sentence has been added to the text: “Further experiments should be performed to define the role of beer's polyphenols in altering SCFA-producing gut microbiota”.
10. The severity of liver degeneration was classified according to the following semi-quantitative scoring system: 0, <5% of hepatocytes affected by fat vacuoles; 0.5, fat vacuoles seen in 5–15% of hepatocytes; 1, fat vacuoles seen in 15–30% of hepatocytes; 2, 30–60% of hepatocytes affected by fat vacuoles; 3, fat vacuoles seen in 60–80% of hepatocytes; 4, <80% of hepatocytes affected by fat vacuoles. The system proposed by the NASH Clinical Research Network was used [a)], slightly modified following b).
- a) Kleiner DE, Brunt EM, Van Natta M, Behling C, Contos MJ, Cummings OW, et al., Design and validation of a histological scoring system for nonalcoholic fatty liver disease. Hepatology 41: 1313–1321 (2005).
- b) Pozzo L, Pucci L, Buonamici G, Giorgetti L, Maltinti M, Longo V. Effect of white wheat bread and white wheat bread added with bioactive compounds on hypercholesterolemic and steatotic mice fed a high-fat diet. J Sci Food Agric. 2015 Sep;95(12):2454-61. doi: 10.1002/jsfa.6972. Epub 2014 Nov 24. PMID: 25348650.
Figure S4 (Steatosis scores for mice liver tissues) has been introduced in the Supplementary file. Consequently, the M&M text in section 4.3.5 has been revised, the figure discussed in Results section 2.3.3, and references modified accordingly.
11. In the discussion we have emphasized that findings are specific to the tested beer type and inserted the suggestion of exploring effects across diverse beer formulations.
12. After production by the company, the beers were transported to our facility under controlled temperature conditions and subsequently stored at 4°C until utilized for experimental purposes to ensure consistency in polyphenol content throughout the experiment.
13. The other quantitatively less representative detected phenolic compounds have been briefly discussed in the discussion paragraph.
14. The claim that transcriptomic changes in NAFLD are reversible were reported in previous studies and are described in the introduction section: “Transcriptomic changes in mice following HFD treatment were found to be reversible, with partial reversion occurring after weight loss [23], in other case mice with an antisteatotic treatments showed a reversion of HFD transcriptomic signature, although the extent varied depending on the treatment [24]”. Our experimental design does not allow us to support these claims.
15. We have mentioned potential confounders in in vitro antioxidant assays, highlighting the need for in vivo validation of antioxidant activity.
16. The limitation of using only one beer dose has been added in the discussion, together with the suggestion of further studies with varying doses to identify thresholds for toxicity and efficacy.
17. The multifactorial nature of NAFLD has been emphasized and beer’s role within these mechanisms has been specified in the discussion paragraph.
18. We have elaborated how cecal metabolites, especially succinic acid and propionic acid, impact systemic inflammation, lipid metabolism and gut-liver axis in the discussion paragraph, introducing new references.
19. The insights from the CTR+beer group, such as baseline metabolic changes and adaptive responses have been discussed more in detail.
20.
21. We have specified in the text that he ability of beer in reducing methylation found in our study is liver-specific and we have proposed exploring effects in other organs like adipose tissue.
21. Done
22. Recommendations for at risk groups like pregnant women, individuals with liver diseases, or those on medication have been explicated in the text.
Minor comments
1. The sentence has been rephrased as suggested.
2. “Poorness” has been replaced with “lack”.
3. The sentence has been corrected as suggested.
4. The sentence has been corrected as suggested.
5. The sentence has been corrected as suggested.
6. “individual” has been removed as suggested.
7. “among” has been replaced with “by”.
Others:
1. NAFLD has been changed to Metabolic dysease-Associated Steatotic Liver Disease (MASLD) throughout the text, according to the new nomenclature following the suggestion of reviewer 2.
2. The sentence has been rephrased.
3. HFD has been hyphenated throughout all the text.
4. Revised
5. Corrected
6. Revised
7. Corrected
8. Corrected
9. Revised
10. Done
11. Done
12. Revised
13. Corrected
14. Corrected
15. Revised
Changes in the text are tracked in red.
Reviewer 2 Report
Comments and Suggestions for Authors
In this manuscript, the authors performed in vivo studies for determining the effect of beer intake on MASLD development. The concept is novel, and the authors adequately translated the experimental data. However, there are a few points that need to be addressed to improve the quality of the manuscript.
1. Please change NAFLD to MASLD according to the new nomenclature
2. The current study is largely descriptive, so it is recommended that the authors should perform some mechanistic studies. For example, please perform western blot or RT-qPCR to check the changes in gene expression related to oxidative stress, liver injury, and/or inflammation.
3. Figure 1.
(1) 1A: please place AST in the middle of the graph.
(2) Please describe the statistical analysis method following ANOVA (such as Tukey) for multiple comparison.
(3) Please add the diagram that describes the experimental procedure (feeding of HFD and supply of alcohol).
(4) How was HDL-cholesterol changed by the feeding?
4. Figure 2
(1) There is no panel A in the figure.
(2) Normally, hematoxylin and eosin is abbreviated as “H&E”. Please correct it.
5. Figure 3.
(1) Please remove the red lines under ‘isobutyric acid’, CTR+beer, HFD+beer
Author Response
Please, see the attachment.
